# Activation of Cannabinoid Type 2 Receptor in Microglia Reduces Neuroinflammation through Inhibiting Aerobic Glycolysis to Relieve Hypertension

**DOI:** 10.3390/biom14030333

**Published:** 2024-03-11

**Authors:** Ruohan Shan, Yuxiang Zhang, Yiping Shi, Xiaowen Wang, Xueke Wang, Guanying Ma, Qian Li

**Affiliations:** 1Department of Physiology, Hebei Medical University, Shijiazhuang 050017, China; shanruohan61@163.com (R.S.); zhangyuxiang@stu.hebmu.edu.cn (Y.Z.); shiyiping@stu.hebmu.edu.cn (Y.S.); wangxiaowen@stu.hebmu.edu.cn (X.W.); wangxueke@stu.hebmu.edu.cn (X.W.); guanyingma@stu.hebmu.edu.cn (G.M.); 2Cardiovascular Research Platform, Institute of Medicine and Health, Hebei Medical University, Shijiazhuang 050017, China

**Keywords:** cannabinoid type 2 receptor, microglia, aerobic glycolysis, neuroinflammation, hypertension

## Abstract

Background: Studies have shown that the chronic use of cannabis is associated with a decrease in blood pressure. Our previous studies prove that activating the cannabinoid type 2 (CB2) receptor in the brain can effectively reduce blood pressure in spontaneously hypertensive rats; however, the exact mechanism has not been clarified. The objective of this study is to demonstrate that activation of microglial CB2 receptors can effectively reduce the levels of TNF-α, IL-1β, and IL-6 in the paraventricular nucleus (PVN) through inhibiting aerobic glycolysis, thereby relieving hypertension. Methods: AngiotensinII (AngII) was administered to BV2 cells and C57 mice to induce hypertension and the release of proinflammatory cytokines. The mRNA and protein expression of the CB2 receptor, TNF-α, IL-1β, IL-6, and the PFK and LDHa enzymes were detected using RT-qPCR and Western blotting. The Seahorse XF Energy Metabolism Analyzer was used to measure the oxidative phosphorylation and aerobic glycolysis metabolic pathways in BV2 cells. The long-term effects of injecting JWH133, a selective CB2 receptor agonist, intraperitoneally on blood pressure were ascertained. ELISA was used to measure norepinephrine and lactic acid levels while immunofluorescence labeling was used to locate the CB2 receptor and c-Fos. By injecting pAAV-F4/80-GFP-mir30shRNA (AAV2-r-CB2shRNA) into the lateral cerebral ventricle, the CB2 receptor in microglia was specifically knocked down. Results: Activation of CB2 receptors by the agonist JWH133 suppressed TNF-α, IL-1β, and IL-6 by inhibiting PFK and LDHa enzymes involved in glycolysis, as well as lactic acid accumulation, along with a reduction in glycoPER levels (marks of aerobic glycolysis) in AngII-treated BV2 cells. In AngII-treated mice, the administration of JWH133 specifically activated CB2 receptors on microglia, resulting in decreased expression levels of PFK, LDHa, TNF-α, IL-1β, and IL-6, subsequently leading to a decrease in c-Fos protein expression within PVN neurons as well as reduced norepinephrine levels in plasma, ultimately contributing to blood pressure reduction. Conclusion: The results suggest that activation of the microglia CB2 receptor decreases the neuroinflammation to relieve hypertension; the underlying mechanism is related to inhibiting aerobic glycolysis of microglia.

## 1. Introduction

Cannabinoids, both synthetic and endogenous, have physiological and pathological regulatory effects. The described effects are mediated by G protein-coupled cannabinoid (CB) receptors, specifically the CB1 receptor [1,2,3] and the CB2 receptor [4,5,6]. Cannabis has well-documented effects on the nervous and immune systems, as well as the circulatory system, causing significant hypotension [7,8,9,10]. Chronic cannabis use has been linked to a decrease in blood pressure [11,12,13]. Our previous study has demonstrated that long-term injection of the CB2 receptor agonist JWH133 into the lateral ventricles of spontaneously hypertensive rats effectively reduces blood pressure [14]. However, the mechanism linking cannabinoid receptors and hypotension is not yet fully understood.

The sympathetic cardiovascular regulatory center is primarily composed of the rostral ventrolateral medulla (RVLM) and hypothalamic paraventricular nucleus (PVN) areas. Their activities regulate the heart and blood vessels, thereby affecting blood pressure. Studies have shown that increased activity in these areas is involved in the pathophysiology of hypertension [15,16,17]. Recent studies have shown that neuroinflammation mediated by microglia enhances the activity of the sympathetic cardiovascular center, leading to an increased excitability of sympathetic nerves [18,19].

Microglia are immune cells in the central nervous system that protect against pathogen invasion. When activated, they release high concentrations of proinflammatory cytokines (TNF-α, IL-1β, and IL-6), which can harm the neurological system [20]. Studies by Tan and Wu et al. have shown that proinflammatory cytokines (IL-1β, TNF-α, IL-6) were increased in the RVLM and PVN regions of spontaneous hypertensive rats and stressed hypertensive rats [21,22]. Reducing the expression of proinflammatory cytokines effectively relieves hypertension [14].

The cannabinoid receptor 2 (CB2) receptor exists in peripheral immune cells and the central nervous system, and many reports indicate that the CB2 receptor is highly expressed in activated microglia during neuroinflammation induced by cerebral ischemia and brain injury [6], as well as neurodegenerative diseases [11]. Activating CB2 receptors on microglia significantly reduces neuroinflammation, relieving and treating these diseases [11]. Nevertheless, to date there have not been any studies performed that look into whether specifically activating CB2 receptors on microglia can lower neuroinflammation and treat hypertension. The underlying mechanism is even more elusive.

In this research, BV2 cells and C57 mice were treated with AngII to simulate a hypertension environment. The relationship between the activation of CB2 receptors and the metabolic reprogramming of microglia was investigated. This study reports the first evidence that inhibiting aerobic glycolysis (glycolysis occurring in aerobic states) in microglia via CB2 receptor activation leads to a reduction in proinflammatory cytokine production, which helps alleviate hypertension.

## 2. Materials and Methods

### 2.1. Chemical and Drugs

A solution of 1 mg/mL was obtained by dissolving 1 milligram of lipopolysaccharide (LPS, Cat. No. L2880, Sigma Aldrich, Shanghai, China) powder in 1 milliliter of sterile normal saline. To create a 10 mg/mL AngII storage solution, 5 mg of L-isoleucyl-L-histidyl-L-prolyl-L-phenylalanine and 2,2,2-trifluoroacetate (AngII, Cat. No. 24739, Cayman Chemical, Ann Arbor, MI, USA) powder were dissolved in 500 μL of sterile phosphate-buffered saline (PBS, Cat. No. P1020, Beijing Solarbio Biotechnology Co., Ltd., Beijing, China). Then, 5 mg powder of 3-(1,1-dimethylbutyl)-6aR,7,10,10aR-tetrahydro-6,6,9-trimethyl-6H-dibenzo[b,d]pyran (JWH133, Cat. No. 259869-55-1, ApexBio Technology, Houston, TX, USA) was dissolved in 800 μL of dimethyl sulfoxide (DMSO, Cat. No. D8371, Beijing Solarbio Biotechnology Co., Ltd., Beijing, China). The mixture was heated at 37 °C in a water bath to create a storage solution containing 20 mM JWH133. During the experiment, the final concentration of DMSO was less than 0.1%.

### 2.2. Animal Methods

#### 2.2.1. Animals

Adult male C57 mice (*n* = 30, 8–9 weeks old, weighing roughly 23 g) were bought from Beijing Vital River Laboratory Animal Technology Co., Ltd. (Beijing, China). They were kept in a controlled environment with enough food and water (12:12 h light–dark; temperature maintained at 20 ± 2 °C). All animal procedures were authorized by Hebei Medical University’s Animal Care and Use Committee and followed the Chinese Animal Care Committee’s requirements. The guidelines for the care and use of laboratory animals (Life Sciences Council Laboratory Animal Resources Institute, 2011) were adhered to when handling and feeding the mice.

#### 2.2.2. Brain Stereotaxic Injection

The mice were anesthetized intraperitoneally with ketamine hydrochloride and xylazine (Cat. No. PHR8715, Cat.No.X1126, Sigma-Aldrich, St. Louis, MO, USA) (66.6 and 1.3 mg/kg, respectively). A stereotaxic frame (RWD Life Science Inc., Shenzhen, China) was used to hold the mice’s heads. A glass pipette with a tip diameter of 20–30 μm was inserted into the lateral ventricle (LV) for the LV microinjections, as previously described [14]. This was performed by boring a tiny hole in the dorsal surface of the skull at 0.48 mm caudal to the bregma, 1.00 mm lateral to the midline, and 2.00 mm ventral to the dura. Each mouse received 100,000 U/kg i.p. of penicillin G procaine (Cat. No. 1502552, Sigma-Aldrich, St. Louis, MO, USA) and 0.05 mg/kg s.c. of buprenorphine (Cat. No. PHR8955, Sigma-Aldrich, St. Louis, MO, USA) to alleviate discomfort. For the purpose of gene knockdown, 3 μL pAAV-F4/80-GFP-mir30shRNA (AAV-CB2-shRNA; Weizhen Co., Ltd., Jinan, China; specifically knock down of the CB2 receptor in microglia) or an AAV-control virus was injected into the LV at a speed of 100 nL/min for 15 min. Any other surgeries were performed ten days later. 

#### 2.2.3. Implant Osmotic Pump and Intraperitoneal Injection

An osmotic pump (RWD Life Science Inc., Shenzhen, China) was filled with AngII (500 μg, 200 μL) or PBS (200 μL). Mice treated with the AAV were given the general anesthesia indicated above, and the skin between their shoulder blades and subcutaneous tissue was sliced. After the osmotic pump was inserted into the subcutaneous area, the skin incision was sutured. For a duration of 28 days, the mice were consistently administered AngII or PBS.

JWH133 (2 mg/kg) or control solvent were injected intraperitoneally into mice once a day for 28 days, starting with the osmotic pump implantation.

#### 2.2.4. Blood Pressure Measurement

In conscious mice, a noninvasive tail-cuff device, Rodent NIBP CODA^®^ Monitor (AD instrument, Bella Vista, NSW, Australia) was used to measure the mean arterial pressure (MAP), diastolic blood pressure (DBP), and systolic blood pressure (SBP). The results were obtained by averaging a minimum of six consecutive measurements.

In anesthetized mice, blood pressure (BP) was monitored, as previously described [14]. During the trial, mechanical ventilation was performed through tracheal cannulation using a rodent ventilator (RWD407, Shenzhen, China) with 100% oxygen. Blood pressure was constantly measured by surgically inserting a cannula into the left femoral artery. Blood pressure readings were collected using the Labchart7 system and a power lab (AD instrument, Bella Vista, NSW, Australia).

#### 2.2.5. Measurement of Plasma Norepinephrine

After administering the previously mentioned anesthetic, using EDTA-Na2 (Cat. No. 03690, Sigma-Aldrich, St. Louis, MO, USA) as an anticoagulant, blood samples were taken from the cannula that was inserted into the left femoral artery. The samples were separated into plasma by centrifuging them at 1000 g for 15 min at 4 °C. The norepinephrine ELISA Kit (Cat. No. EU2565, Nanjing Jiancheng Bioengineering Institute, Nanjing, China) instructions were followed for the collection and processing of the plasma. The following were added in order: biotinylated antibody working solution, enzyme binding working solution, color development substrate, and termination solution. The optical density (OD value) at 450 nm wavelength was determined using a microplate reader (Molecular Devices, San Jose, CA, USA). Relying on the acquired standard curve, the plasma norepinephrine concentrations in every group were computed.

#### 2.2.6. Frozen Section and Immunofluorescence Staining

The mice were anesthetized as described above and infused with 4% paraformaldehyde (PFA, Cat. No. P1110, Beijing Solarbio Biotechnology Co., Ltd., Beijing, China) through the left ventricle. The brains were successively placed in 4% PFA and then dehydrated in the 30% sucrose solution for 48 h. After cryoprotection, the frozen brain coronal sections, including the PVN, were made into 20 μm thick sections using a freezing microtome (CM1950, Leica Biosystems, Wetzlar, Germany); then, immunofluorescence staining was performed.

The primary antibodies included CB2 antibody (Cat. No. DF8646, Affinity Biosciences, Cincinnati, OH, USA) at 1:500, Iba1 antibody (Cat. No. ab153696, Abcam, Cambridge, MA, USA) at 1:400, and c-Fos antibody (Cat. No. ab214672, Abcam, Cambridge, MA, USA) at 1:1000. The appropriate proportion of primary antibody was added to 0.01 M PBS (Cat. No. KGB5001, Kaiji, Nanjing, China) containing 5% BSA (Cat. No. SW3015, Beijing Solarbio Biotechnology Co., Ltd., Beijing, China) and 0.25% Triton X 100 (Cat. No. T8200, Beijing Solarbio Biotechnology Co., Ltd., Beijing, China), and was treated at 4 °C overnight (16–18 h). The slices were then incubated in a mixture of fluorescent secondary antibodies DyLight405/DyLight594-conjugated anti-mouse/anti-rabbit IgG (Report Biotechnology Co., Ltd., Shijiazhuang, China) at 1:500 at room temperature for 2 h. All images were obtained using fluorescence microscopy (DM600, Leica Biosystems, Wetzlar, Germany).

### 2.3. Cell Culture Methods

#### 2.3.1. Cell Culture

The immortalized murine microglial cell line BV2 was purchased from Procell Life Science and Technology Co., Ltd. BV2 cells were incubated in MEM complete medium (Cat. No. PM150478, Procell Life Science and Technology Co., Ltd., Wuhan, China) at 37 °C with 5% CO_2_. Once the cell growth density reached about 80%, the subsequent experiment was carried out. 

To knockdown the CB2 receptor on BV2 cells, complete medium without penicillin-streptomycin was supplemented with CB2-siRNA (GATCCCTAACGACTACCTA) (50 nM) or control SiRNA (TGAGGTTGGCCAAGACTCT) and transfection reagent (riboFECTTMCP Reagent, Cat.No.C10511-05, Ribobio Technology, Guangzhou, China) for 48 h. Then, the complete media were replaced to carry out the next experiments.

The cells were treated for 36 h with either the control solvent or AngII (the final concentration of 100 nM). Before AngII was administered to the cells for 12 h, the CB2 receptor agonist JWH133 (at a final concentration of 100 nM) or the control solvent was added, so that the total action time of JWH133 was 48 h. 

#### 2.3.2. Cell Proliferation Activity and Cell Morphology

Cell Counting Kit-8 (CCK8, Cat. No. K1018, Apex Bio, Houston, TX, USA) was used to measure the activity of cell proliferation. As directed by the manufacturer, BV2 cells were cultivated in 96-well plates. After that, the CCK8 detection solution was added and incubated for 30 min. A microplate reader (Molecular Devices, San Jose, CA, USA) was used to measure the optical density at 450 nm.

BV2 cells were cultured in 6-well plates and subsequently observed under a microscope (DMi8, Leica Biosystems, Wetzlar, Germany) to assess the morphological alterations within each experimental group.

#### 2.3.3. Lactic Acid Detection

The supernatant and cell lysate of the cell culture medium were collected. By using the Lactic Acid Detection Kit (Cat. No. A019-2, Nanjing Jiancheng Bioengineering Institute, Nanjing, China) protocol, six standard concentrations were prepared to create a standard curve, and the samples were diluted for measurement. The enzyme working liquid and chromogenic agent were added to react in the 37 °C water bath for 10 min, and then an appropriate volume of termination fluid was added to end the reaction. The OD value was detected using a 530 nm microplate reader (Molecular Devices, San Jose, CA, USA). Finally, the content of lactic acid in the sample was calculated using the formula for lactic acid calculation.

#### 2.3.4. Cell Aerobic Glycolysis Rate

The Seahorse XF Energy Metabolism Analyzer (Agilent, Santa Clara, CA, USA) was utilized for real-time measurement of oxidative phosphorylation and aerobic glycolysis metabolic pathways in the live cells to quantify cell metabolic phenotypes. The oxygen consumption rate (OCR) reflected the mitochondrial function, and the extracellular acidification rate (ECAR) reflected the glycolysis function. To obtain the proton outflow rate during glycolysis (glycoPER), the acidification rate contributed by CO_2_ was subtracted from the total proton outflow rate (PER), resulting in glycoPER which was discharged into the extracellular solution. As per the Seahorse XF Glycolytic Rate Assay Kit (Cat. No. 103346-100, Agilent, Santa Clara, CA, USA), BV2 cells that had been pretreated with CB2-siRNA, AngII, and JWH133 were seeded in poly-Lysine-coated XFe 96-well seahorse culture microplates at a concentration of 50 × 10^4^ cells per well. The plates were then incubated overnight at 37 °C in 5% CO_2_ prior to the treatment. After following the prescribed treatments for 24 h, 175 μL of pre-warmed bicarbonate-free DMEM (basal) assay medium was added to the growing media, which was then aspirated and cleaned thrice. In order to degas the plate, it was incubated for 45 min at 37 °C in a CO_2_-free environment. Following incubation, OCR and ECAR were measured according to the manufacturer’s protocol with the following chemicals present: 50 mM 2-DG and 0.5 μM Rot/AA.

### 2.4. Western Blot

The BV2 cells were removed from the original medium and washed with pancreatic enzyme (Cat. No. T1350, Beijing Solarbio Science and Technology Co., Ltd., Beijing, China). The cells were collected, and protein extraction was performed by adding lysate (Cat. No. P0013, Beyotime Biotechnology Co., Ltd., Shanghai, China) and protease inhibitors (Cat. No. P1005, Beyotime Biotechnology Co., Ltd., Shanghai, China). The mice were euthanized via quick decapitation under deep anesthesia; the brains were dissected out, and the PVN tissues were collected using the punch microdissection technique as described in the previous report [14]. Punched tissue was kept at −80 °C in the lysis solution containing protease inhibitors.

The protein concentrations in the samples were determined by using the BCA method (Cat. No. RW0201, Report Biotechnology Co., Ltd., Shijiazhuang, China), and then the protein samples were prepared. The proteins underwent electrophoresis in the 10% or 12% TRIS-HCL (Cat. No. RS0043, Report Biotechnology Co., Ltd., Shijiazhuang, China) polyacrylamide gel. Following electrophoresis, the proteins were transferred onto PVDF membranes (Millipore, Burlington, MA, USA). After the membrane transfer, the PVDF membrane was put into the sealing solution containing pre-prepared 5% skim milk powder (Cat. No. P0216, Beyotime Biotechnology Co., Ltd., Shanghai, China) and incubated at room temperature for 1 h with agitation. Next, the membranes were exposed to diluted primary antibodies overnight at 4 °C: TNF-α antibody (Cat. No. AF7014, Affinity Biosciences, Cincinnati, OH, USA) at 1:2000, IL-1β antibody (Cat. No. AF5103, Affinity Biosciences, Cincinnati, OH, USA) at 1:1000, IL-6 antibody (Cat. No. DF6087, Affinity Biosciences, Cincinnati, OH, USA) at 1:500, CB2 antibody (Cat.No.DF8646, Affinity Biosciences, Cincinnati, OH, USA) at 1:2000, PFK antibody (Cat.No.AF7562, Affinity Biosciences, Cincinnati, OH, USA) at 1:500, LDHa antibody (Cat. No. DF6280, Affinity Biosciences, Cincinnati, OH, USA) at 1:1000, β-actin antibody (Cat. No. AF0003, Beyotime Biotechnology Co., Ltd., Shanghai, China) at 1:500. Then, the members were incubated with secondary antibodies at 37 °C for 1 h, with goat anti-rabbit IgG (Cat. No. A0208, Beyotime Biotechnology Co., Ltd., Shanghai, China) used at 1:10,000. Electrochemical luminescence (ECL, Cat. No. P0018S, Beyotime Biotechnology Co., Ltd., Shanghai, China) was employed for signal development, and ImageJ software (X64, V1.8.0) was used to analyze the gray values obtained from the bands. The gray value of the target protein in each group was normalized to the gray value of the internal parameter β-actin. The ratio of each group was further normalized by the control group. 

### 2.5. Total RNA Extraction and RT-qPCR

The cultured cells and the brain PVN tissues were collected just as described above. According to the protocol, the total RNA was extracted using the Trizol Kit (Cat. No. TR201-50, Beijing Tianmo Sci and Tech Development Co., Ltd., Beijing, China), followed by reverse transcription of the total RNA into cDNA using pre-designed primers (TNFα, IL-1β, IL-6, CB2, and GAPDH; the sequences are listed in the Appendix A and were purchased from Sangon Biotech Co., Ltd. (Shanghai, China) and the SYBR Green PCR Kit (Cat. No. MQ10201, Monadbiotech Co., Ltd., Suzhou, China). The threshold cycle number was determined by triplicate analysis of each sample. For normalization purposes, GAPDH mRNA was utilized as an internal control. The relative expression levels were calculated using the comparative 2^−∆∆CT^ method and reported as fold changes relative to the controls. 

### 2.6. Statistical Analyses

The experimental results were analyzed using GraphPad Prism 8 statistical software. The data are expressed as mean ± SEM. Two-way or one-way ANOVA were used for data analysis. Tukey post hoc tests were conducted to investigate significant main and interaction effects when appropriate. A statistically significant difference was indicated by *p* < 0.05. 

## 3. Results

### 3.1. In AngII-treated Mice, Activation of CB2 Receptors by JWH133 Decreased the Blood Pressure, Plasma Norepinephrinein, c-Fos Expression, Proinflammatory Cytokine, and Enzyme Associated with Glycolysis in the PVN

After 28 days of continuous AngII administration, the CB2 protein (3.29 ± 0.46) was highly increased in the PVN. Moreover, the CB2 protein (dyed red) was found to be expressed in microglia (Iba1 stained blue) by using immunofluorescence staining. AAV-CB2-shRNA was used to target microglia specifically, and the resultant knockdown of the CB2 receptor led to a significant decrease in CB2 receptor protein levels (0.21 ± 0.02) in the PVN, as well as a reduction in CB2 receptors within microglia cells specifically (Figure 1a). 

There was a notable rise in MAP in both conscious (138.01 ± 2.01 vs. 103.72 ± 2.44) and anaesthetized (133.92 ± 1.65 vs. 112.23 ± 1.33) AngII-treated animals (Figure 1b). Simultaneously, the plasma norepinephrine level increased (Figure 1c): 22.37 ± 1.35 in AngII-treated vs. 5.41 ± 0.50 in Sham. The PVN of mice treated with AngII had higher levels of TNF-α (2.28 ± 0.13), IL-1β (3.12 ± 0.25), and IL-6 (2.54 ± 0.13) (Figure 1d); the glycolysis-related enzymes, PFK (2.35 ± 0.21) and LDHa (2.76 ± 0.20), were similarly elevated (Figure 1e).

The CB2 receptor agonist JWH133 (2 mg/kg) was injected intraperitoneally for 28 days into the AngII-treated mice. As a result, the MAP, norepinephrine, TNF-α, IL-1β, IL-6, PFK, and LDHa levels in the AngII-treated mice were all decreased to that of the Sham group. Knocking down the microglia CB2 receptor did not specifically relieve the enhancement of MAP, norepinephrine, TNF-α, IL-1β, IL-6, PFK, and LDHa induced by AngII, but it significantly reversed the inhibition effects caused by JWH133. (Figure 1b–e)

In the PVN of AngII-treated mice, immunofluorescent staining demonstrated a significant increase in the c-Fos protein expression (stained in red), indicating neuronal excitation. After the administration of JWH133, the c-Fos levels in mice treated with AngII decreased. Although knocking down the microglia CB2 receptor did not specifically alleviate the enhancement of c-Fos induced by AngII, it did significantly reverse the inhibitory effects caused by JWH133. Additionally, there was no alteration in Iba1-positive microglia (stained in blue), and no colocalization between c-Fos and microglia was seen. (Figure 2)**.**

### 3.2. The BV2 Cells Treated with AngII Exhibited an Increase in Proinflammatory Cytokines

The cell viability was assessed, and the results showed that following 24 h of treatment with 1 μg/mL LPS, the proliferation level of BV2 cells decreased from (1.00 ± 0.03) to (0.90 ± 0.01) *(p* < 0.05). While the BV2 cells were treated with AngII (100 nM, 36 h), the proliferation (1.22 ± 0.03) was higher than that in control group (1.00 ± 0.09) (*p* < 0.05) (Figure 3a). 

The morphology of the BV2 cells was observed under a microscope. The control cells were mostly fusiform or regular round, and the synapses were branching. After 24 h treatment with 1 μg/mL LPS, the synapses of the BV2 cells became thicker and shorter, from branching to amoebic, and the cell volume increased, which is the same morphology change as the BV2 cells treated with 100 nM AngII for 36 h (Figure 3c).

The mRNA expressions of TNF-α (40.64 ± 0.80), IL-1β (726.87 ± 40.12), and IL-6 (60.07 ± 0.84) were all significantly increased in the BV2 cells treated with LPS (1 μg/mL, 24 h). Similarly, the mRNA expression of TNF-α (3.16 ± 0.03), IL-1β (13.56 ± 0.14), and IL-6 (1.51 ± 0.08) were also all significantly increased in the BV2 cells treated with AngII (100 nM, 36 h) (Figure 3b). At the same time, the protein levels of TNF-α (1.48 ± 0.06), IL-1β (1.53 ± 0.04), and IL-6 (1.92 ± 0.12) were all significantly increased in the BV2 cells treated with AngII (100 nM, 36 h) (Figure 3d).

### 3.3. An Increase in Proinflammatory Cytokines Caused by AngII in BV2 Cells Was Reduced by Activating CB2 Receptors with the Agonist JWH133

Compared with the control group (1.00 ± 0.13), the protein expression of the CB2 receptor (3.29 ± 0.46) was significantly increased in the BV2 cells treated by AngII (100 nM, 36 h) (Figure 4a). Using CB2-siRNA (GATCCCTAACGACTACCTA, Rui Bo Biological Co., Ltd., Shanghai, China) (50 nM, 48 h) resulted in a 73% reduction in CB2 receptor protein expression compared to the control group (Figure 4b). Activating the CB2 receptors in BV2 cells using the agonist JWH133 (100 nM, 48 h) inhibited the TNF-α, IL-1β, and IL-6 protein elevations induced by AngII. Knocking down the CB2 receptors in BV2 cells promoted the elevation of AngII-induced TNF-α, IL-1β, and IL-6, and reversed the inhibitory effects of JWH133 (Figure 4c).

### 3.4. The Increase in Aerobic Glycolysis Produced by AngII in BV2 Cells Was Reduced by Activating the CB2 Receptor with the Agonist JWH133

The lactic acid test results revealed that the lactic acid content in the BV2 cells treated with AngII was significantly increased compared to the control group, both in the supernatant (11.00 ± 0.12) and in cell lysate (0.60 ± 0.01), as opposed to the control group’s supernatant of cell culture medium (5.10 ± 0.12) and cell lysate (0.51 ± 0.01). However, when the CB2 receptors in BV2 cells were activated by JWH133 agonist treatment (100 nM, 48 h), it effectively inhibited the elevation of lactic acid content induced by AngII, both in the supernatant (5.95 ± 0.13) and in the cell lysate (0.46 ± 0.00). On the other hand, knocking down the CB2 receptors in BV2 cells further increased AngII-induced lactic acid content in both the supernatant (14.39 ± 0.28) and cell lysate (0.71 ± 0.01) while also reversing JWH133’s inhibitory effects on lactic acid levels, as observed in the supernatant (14.65 ± 0.33) and cell lysate (0.53 ± 0.03) (Figure 5a).

Enzymes associated with glycolysis (PFK and LDHa) were also observed. The protein expressions of PFK and LDHa were significantly increased in the BV2 cells treated with AngII. Activation of the CB2 receptors in BV2 cells by agonist JWH133 (100 nM, 48 h) inhibited the PFK and LDHa elevations induced by AngII. Knocking down the CB2 receptors in BV2 cells promoted the elevation of AngII-induced PFK and LDHa levels while reversing the inhibitory effects of JWH133 on PFK and LDHa (Figure 5b).

The Seahorse XF Energy Metabolism Analyzer was utilized to quantitatively assess the metabolic phenotypes of live cells by measuring real-time aerobic glycolysis metabolism. The results showed that compared with the control group (35.10 ± 2.03), the glycoPER value (69.21 ± 2.20) increased in the BV2 cells treated with AngII and subsequently decreased to 34.19 ± 1.11 after JWH133 treatment. Furthermore, knocking down CB2 receptors resulted in a higher glycoPER value (154.30 ± 13.60) compared to the AngII group and reversed the inhibitory effect of JWH133 on glycoPER (158.60 ± 6.90) (Figure 5c).

### 3.5. CB2 Receptor Agonist JWH133 Reversed Proinflammatory Cytokine Increase through Inhibiting Aerobic Glycolysis Enhancement Induced by AngII

In the same results as above, the lactic acid in the supernatant of the cell culture medium and cell lysate increased significantly in the BV2 cells treated with AngII, while JWH133 reversed the AngII-induced lactic acid increase (Figure 6a). Oligomycin, as an ATP synthase inhibitor, greatly improved the glycolysis level of the cells [23]. The results show that oligomycin (2.5 μg/mL, 48 h) further enhanced the lactic acid in the supernatant (16.02 ± 0.25) and cell lysate (3.14 ± 0.45). Moreover, oligomycin effectively reversed the inhibitory effect of JWH133 on AngII-induced lactic acid increase. Additionally, oligomycin significantly increased the protein levels of TNF-α (2.50 ± 0.18), IL-1β (2.70 ± 0.24), and IL-6 (2.45 ± 0.21). Furthermore, oligomycin effectively reversed the inhibitory effects of JWH133 on TNF-α, IL-1β, and IL-6 increases induced by AngII (Figure 6b–d).

## 4. Discussion

This study provides evidence that glycolysis is significantly increased in PVN microglia in AngII-induced hypertension. Activating CB2 receptors in microglia can suppress glycolysis, which reduces inflammation in the cardiovascular center and lowers blood pressure.

Our study found that MAP and plasma norepinephrine levels were significantly higher in Ang II-treated mice, which is consistent with previous research [24,25,26], indicating successful hypertension modeling. Additionally, TNF-α, IL-1β, and IL-6 levels were significantly elevated in the PVN, indicating neuronal activation and increased excitability. c-Fos protein expression was also elevated in the PVN. The results are consistent with the previous study [18,19]. These findings support the idea that hypertension is accompanied by extensive neuroinflammation in the cardiovascular center. The results of our cellular experiments show that the BV2 cells treated with AngII released more TNF-α, IL-1β, and IL-6, indicating that neuroinflammation associated with microglia accompanies hypertensive PVN hyperactivity.

Microglia are the primary resident immune cells of the central nervous system (CNS). They protrude and allow for real-time observation of the local microenvironment. Normally, they respond to neuronal damage and remove damaged cells by phagocytosis [27,28]. However, in many CNS diseases, such as infections, localized injuries, and hypertension, microglia hyperactivation occurs. This leads to the secretion of more inflammatory mediators, inducing neuroinflammation [29,30,31,32]. How can microglia overactivation be inhibited? Recent studies have shown that in neurological diseases associated with neuroinflammation, such as Alzheimer’s disease and multiple sclerosis, the expression level of the CB2 receptor in activated microglia is significantly elevated. Activation of the CB2 receptor can inhibit microglia overactivation and reduce the release of pro-inflammatory cytokines [6,33]. Furthermore, our previous study shows that the activation of CB2 receptors by JWH133 resulted in a decrease in blood pressure, heart rate, and renal nerve discharge in SHR rats. Additionally, it led to a decrease in TNF-α, IL-1β, and IL-6 in the RVLM [14]. The present animal experimental findings are consistent with our previous studies. Additionally, our present study used AAV-CB2-shRNA to specifically knock down microglial CB2 receptors in Ang II-induced hypertensive mice. This confirmed that the specific activation of microglial CB2 receptors reduces the expression of TNF-α, IL-1β, and IL-6 in the PVN. In addition, our cellular experiments showed that using the agonist JWH133 to activate CB2 receptors inhibited TNF-α, IL-1β, and IL-6 in AngII-treated BV2 cells. The results of the in vivo and in vitro experiments provide evidence to support the idea that the administration of JWH133, which activates microglial CB2 receptors, reduces TNF-α, IL-1β, and IL-6 levels, ultimately leading to a decrease in blood pressure in angiotensinII-induced hypertensive mice.

What is the underlying mechanism of microglia CB2 receptors activation? Jaehong Kim has reported that microglia activation is accompanied by the reprogramming of energy metabolism [34]. In the normal physiological microenvironment, the metabolic pathway of microglia is mitochondrial oxidative phosphorylation [35]. However, in diseases, activated microglia use aerobic glycolysis, which enables them to produce ATP at a faster rate [36]. This results in rapid cell metabolism and the production of more cytokines and reactive oxygen species [37]. Our current results indicate that in BV2 cells treated with AngII, the glycolysis product lactic acid, as well as the glycolysis-related enzymes PFK and LDHa and the metabolic indicators ECAR, PER, and glycoPER, all increased. These findings suggest that aerobic glycolysis is the primary metabolic pathway in AngII-treated BV2 cells. Additionally, our research demonstrates that the activation of CB2 receptors by JWH133 reduces aerobic glycolysis in AngII-treated BV2 cells. To investigate the relationship between aerobic glycolysis and inflammatory factors, we used oligomycin, an ATP synthase inhibitor known to enhance glycolytic activity [23]. Our results showed that oligomycin reversed the inhibitory effects of JWH133 on lactic acid production, TNF-α, IL-1β, and IL-6 release in AngII-treated BV2 cells. Collectively, these findings suggest that targeting CB2 receptors is a promising strategy for suppressing proinflammatory microglial responses by modulating aerobic glycolysis.

There are various restrictions in our investigation. Firstly, we only evaluated the expression of the c-Fos protein in the PVN as a measure of elevated PVN excitability in the cardiovascular central. If experimental conditions permit, it would be beneficial to determine whether c-Fos is expressed in pre-sympathetic neurons and to measure renal nerve discharge and PVN brain slice discharge. Secondly, while CB2 receptor knockdown was performed in microglia throughout the brain, potential effects on other cardiovascular brain regions cannot be ruled out. Additionally, the administration of JWH133 via systemic intraperitoneal injection does not exclude peripheral hypotensive effects. It is important to note that only JWH133 was selected as the agonist in this experiment. To further prove the effect of the CB2 receptor, additional CB2 agonists should be used. Finally, further investigation is required to fully understand the mitochondrial oxidative phosphorylation pathway. Additionally, the signal transduction between the CB2 receptor and aerobic glycolysis enzymes is unclear.

Next, we will apply additional CB2 receptor agonists to activate the receptors and confirm their inhibitory effect on aerobic glycolysis. We will also observe the correlation between CB2 receptor activation and mitochondrial oxidative phosphorylation. Additionally, we will investigate the signaling pathways involved in CB2 receptor activation and observe the signaling molecules that can affect the transcription of enzymes related to aerobic glycolysis. In conclusion, our findings offer new targets and ideas for the clinical treatment of hypertension associated with neuroinflammation.

## Figures and Tables

**Figure 1 biomolecules-14-00333-f001:**
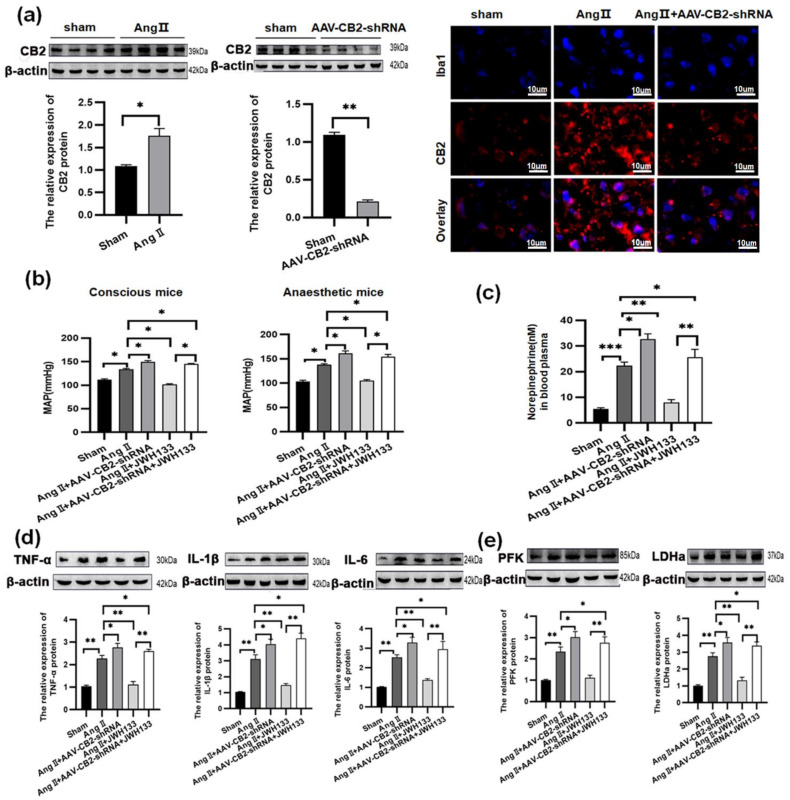
JWH133 (2 mg/kg, 28 days) activated the CB2 receptor on microglia to inhibit AngII-induced hypertension by suppressing glycolytic enzymes and proinflammatory cytokines in the PVN. (**a**) The expression of the CB2 receptor in microglia was upregulated in the PVN of AngII-induced hypertension mice. Iba1 (blue) and CB2 receptor (red). Scale bar = 10 μm. JWH133 activation of the CB2 receptor inhibited MAP (**b**), plasma norepinephrine levels (**c**), proinflammatory factors TNF-α, IL-1β, and IL-6 production (**d**), glycolytic enzymes PFK and LDHa expression (**e**) in AngII-induced hypertension mice. The results are expressed as mean ±SEM (*n* = 5 mice in each group, * *p* < 0.05, ** *p* < 0.01, *** *p* < 0.001). Original blot images can be found in Appendix A.

**Figure 2 biomolecules-14-00333-f002:**
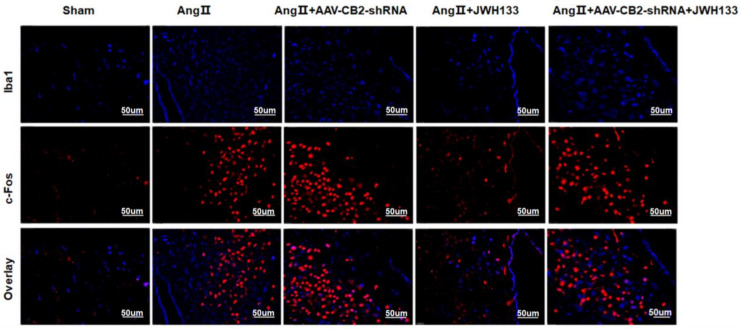
Immunofluorescent staining revealed the expression of the c-Fos protein, which is indicative of neuronal excitation. Notably, no colocalization was observed between c-Fos (stained in red) and microglia (stained in blue). Scale bar = 50 μm.

**Figure 3 biomolecules-14-00333-f003:**
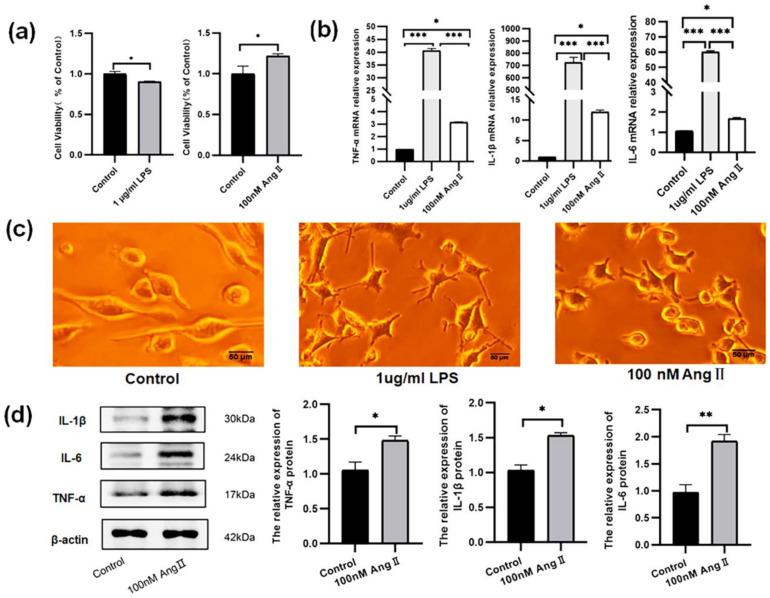
Effect of LPS and AngII on proinflammatory cytokines (TNF-α, IL-1β, and IL-6), morphology, and proliferation of BV2 cells. (**a**) The effects of LPS (1 μg/mL, 24 h) and AngII (100 nM, 36 h) on cell viability in BV2 cells; (**b**) the mRNA expression of TNF-α, IL-1β, and IL-6 in BV2 cells treated with LPS and AngII; (**c**) the morphology of BV2 cells treated with LPS and AngII; (**d**) the original graph and relative protein expression of TNF-α, IL-1β, and IL-6 in BV2 cells treated with AngII. The data are expressed as the mean ± SEM (*n* = 6 times, * *p* < 0.05, ** *p* < 0.01, *** *p* < 0.001). Scale bar = 50 μm. Original blot images can be found in Appendix A.

**Figure 4 biomolecules-14-00333-f004:**
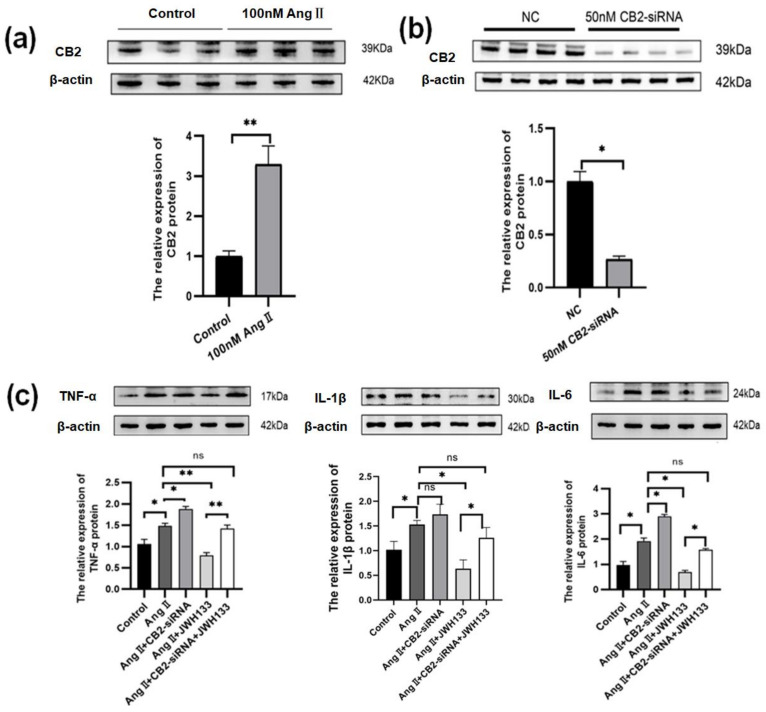
Activation of CB2 receptors by JWH133 inhibited AngII-induced proinflammatory cytokine elevation in BV2 Cells. (**a**) The original graph and protein relative expression of CB2 receptors in BV2 cells treated with AngII; (**b**) the original graph and protein relative expression of CB2 receptors in BV2 cells treated with CB2-siRNA and control-siRNA; (**c**) the original graph and protein relative expressions of TNF-α, IL-1β, and IL-6 in the BV2 cells treated with AngII along with knocking down or activating the CB2 receptor. The data are expressed as the mean ± SEM (*n* = 6 times, * *p* < 0.05, ** *p* < 0.01, ns: *p* > 0.05). Original blot images can be found in Appendix A.

**Figure 5 biomolecules-14-00333-f005:**
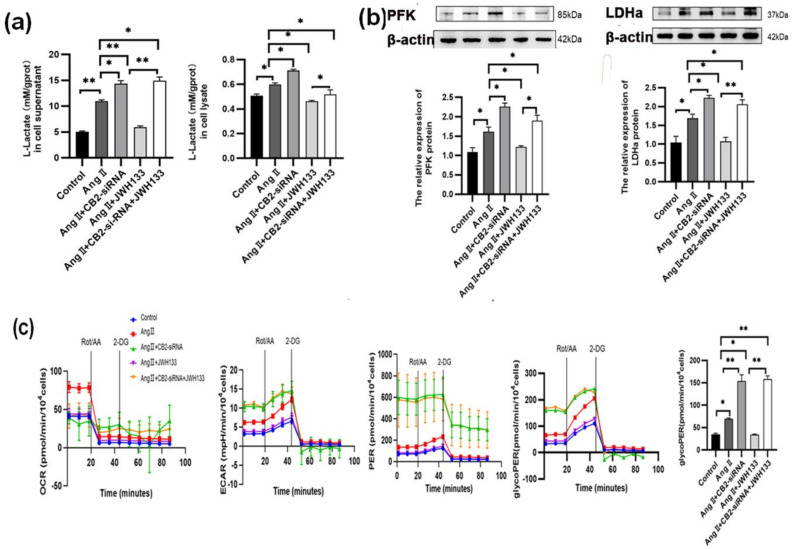
Activation of the CB2 receptor by the agonist JWH133 inhibited AngII-induced aerobic glycolysis augmentation in BV2 cells. (**a**) JWH133 (100 nM, 48 h) activated the CB2 receptor to inhibit AngII-induced lactic acid content in BV2 cell supernatant and cell lysate; (**b**) JWH133 activated the CB2 receptor to inhibit the elevation of PFK and LDHa protein in BV2 cells induced by AngII; (**c**) the activation of the CB2 receptor by JWH133 inhibited the increases in OCR, ECAR, PER, and glycoPER in BV2 cells induced by AngII. The results are expressed as the mean ± SEM (*n* = 5 times, * *p* < 0.05, ** *p* < 0.01). Original blot images can be found in Appendix A.

**Figure 6 biomolecules-14-00333-f006:**
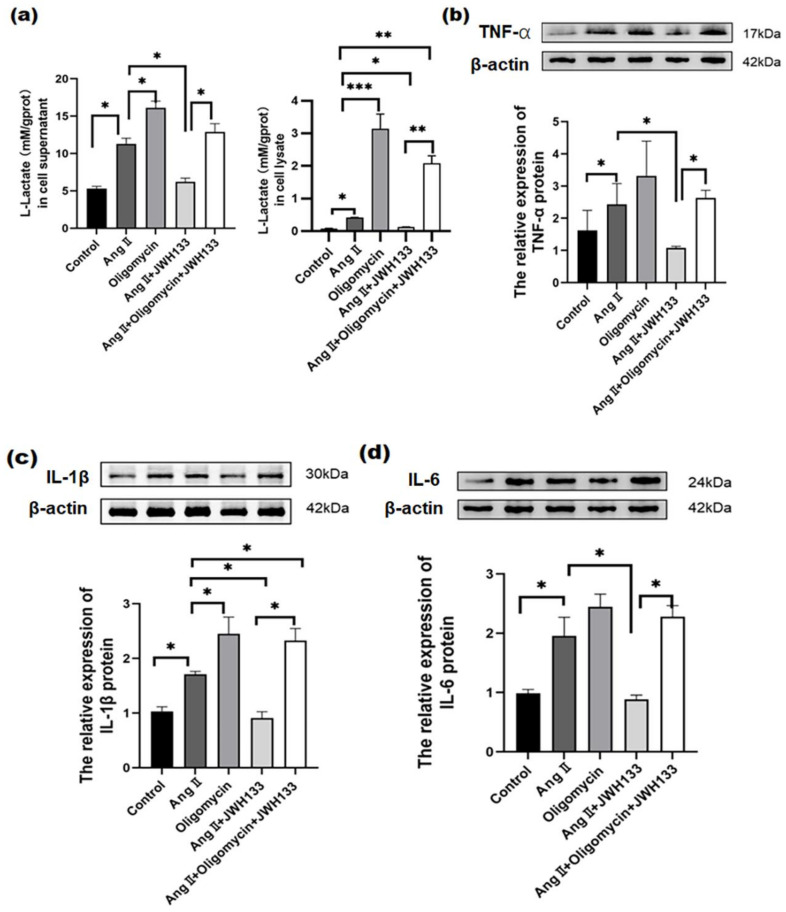
ATP synthase inhibitor oligomycin reversed the inhibitory effects of JWH133 on lactic acid, TNF-α, IL-1β, and IL-6 in BV2 cells treated with AngII. (**a**) Oligomycin effectively counteracted the inhibition of JWH133 on AngII-induced lactic acid increase. Oligomycin effectively reversed the JWH133-mediated inhibition on TNF-α (**b**), IL-1β (**c**), and IL-6 (**d**) protein increases induced by AngII. The results are expressed as the mean ± SEM (*n* = 5 times, * *p* < 0.05, ** *p* < 0.01, *** *p* < 0.001). Original blot images can be found in Appendix A.

## Data Availability

The data supporting this study are available from the corresponding authors upon reasonable request.

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
