# Peer review of "Activation of Cannabinoid Type 2 Receptor in Microglia Reduces Neuroinflammation through Inhibiting Aerobic Glycolysis to Relieve Hypertension"

_biomolecules, 2024, doi:10.3390/biom14030333_

Round 1

Reviewer 1 Report

Comments and Suggestions for Authors

I have reviewed the manuscript of the present investigation which shows evidence of the decrease of proinflammatory cytokines due to the activation of CB2 receptors of microglia, they demonstrate that the decrease of cytokines is due to the fact that in some way the CB2 receptor inhibits aerobic glycolysis. This knowledge is interesting because in the end it is observed that the activation of the CB2 receptor decreases hypertension, so this receptor can be a pharmacological target.

The authors should be careful in the quality of the figures, Figure 1 is the one with the lowest quality and its size is very small, they should consider presenting this figure in two parts, i.e., the photographs of the expression of c-fos could be another figure, the size does not allow to appreciate the results.

In general, the results are interesting and agree with what was observed in the mice with the results of the cell line, I only consider that the authors should make an effort to try to explain or give a hypothesis of how the transductional mechanism of the CB2 receptor inhibits aerobic glycolysis and consequently the synthesis of cytokines.

Finally, the authors should correct references 15 and 23 both of which correspond to the same article.

Comments on the Quality of English Language

The English wording has some errors, needs minor revision.

Author Response

Dear Reviewer:

I hope you have a nice day.

Thank you for your helpful comments on improving the manuscript's quality. We have thoroughly revised the manuscript, correcting minor errors in English wording, grammar, punctuation, and formatting.

We made the following changes in response to specific questions:

  1. The images were resized to improve clarity. Please refer to the revised manuscript.
  2. Figure 1 was divided into two pictures as per your suggestion. Please see the revised Figure 1 and Figure 2.
  3. Additionally, we have corrected the inaccurate references.
  4. Your question about signaling pathways is important, as it helps us understand changes in cells after CB2 receptors are activated. We have reviewed literature to explain the relationship between the CB2 receptor and glycolysis, but due to time constraints, we have not been able to do this work in depth. However, we are happy to discuss relevant findings with you. We have listed possible ways through a literature review below. We hope to receive your guidance and assistance in our upcoming work.

The CB2 receptor is a G-protein-coupled receptor that interacts with the Gα subunit of the Gαi/o type. This interaction inhibits the production of cAMP by AC, resulting in reduced activation of PKA. PKA can regulate the activation of many transcription factors. Therefore, we believe that activating CB2 receptors can inhibit PKA activation and the transcription factors of glycolysis-related enzymes, reducing their transcription and inhibiting glycolysis.

Some literature reports that the PI3K/Akt pathway is involved in signal transduction after CB2 receptor activation and may be involved in glucose metabolism.

Additionally, literature reports show that B-arrestin2 and Erk1/2 molecules act as effector molecules when CB2 receptors are activated. They may also regulate the cellular glucose metabolism.

We believe that activating CB2 receptors leads to complex intracellular molecular changes that involve multiple pathways and effects. It is important to avoid speculation and rely on concrete evidence. Further experiments are needed to observe whether the mentioned molecules have changed and whether blocking these changes can alter the experimental results.

Best Wishes

Reviewer 2 Report

Comments and Suggestions for Authors

This is an interesting manuscript. However, I have some comments and suggestions which may improve the quality of this paper:

Introduction

  1. The introduction is quite long and dense. Consider breaking it up into shorter paragraphs with clear topic sentences to improve readability and help guide the reader.
  2. Be more concise - some details could be trimmed or consolidated to keep the focus on the key points. For example, the details on past research about cannabinoids' effects could be shortened.
  3. Define key terms and concepts. For a general audience, explain what things like "endocannabinoid system", "rostral ventrolateral medulla (RVLM)", and "aerobic glycolysis" are.
  4. Clarify the gap in the research that your study aims to address. What specific piece of the mechanism between CB2 receptors and hypertension is missing or unknown that your study uncovered?
  5. The overall flow and logical progression of ideas could be improved. Group related ideas into coherent paragraphs. Lead with the broad background and funnel down into how your specific study builds on past work.
  6. Proofread to fix minor grammar and style issues. For example, keep verb tenses consistent, avoid overusing words like "show" when introducing cited research, etc.

Methods

  1. The overall structure and organization could be clearer. Consider grouping related methods into subsections, such as "Animal Methods", "Cell Culture Methods", "Immunostaining Methods", etc.
  2. More details are needed in some areas. For example, provide specifics on the stereotaxic surgical procedures, the sequences and suppliers of primers used for qPCR, and the catalog numbers/dilutions of antibodies used for immunoblotting.
  3. Reduce redundancy. Some explanations, like the details of blood pressure measurement, are repeated. Streamline these methods by referring back to their initial explanation.
  4. Standardize terminology and formatting. For example, use either "Anti-Iba1" or "Iba1 antibody", but be consistent. Also ensure consistency in elements like subunit formatting (e.g. all reagents listed as "Cat. No." or "Catalog Number").
  5. Improve readability by breaking lengthy paragraphs into shorter chunks. Add more headings to guide the reader through the procedural explanations.
  6. Carefully proofread for minor errors in grammar, punctuation, spelling and style. For example, ensure consistent verb tense, capitalization of reagents/kits, missing words, etc.
  7. Provide more detail on data analysis. Elaborate on the specific statistical tests used and what variables were compared.

Results

  1. Add more context and explanation for readers not deeply familiar with the field. Elaborate on the significance of findings like blood pressure changes, microglial activation states, etc.
  2. Reduce redundancy between the figure legends and main text. Legends should serve as a complement without excessive repetition.
  3. Improve clarity in areas. For example, explain if conscious vs anesthetized blood pressure measurements serve different purposes. Define any abbreviations upon first use.
  4. Standardize style and formatting for data presentation. For example, use consistent significant digit rules, decimal places, capitalization, etc. when reporting statistical values.
  5. Strengthen connections between experiments. Further discuss how findings in the animal models relate to and support the in vitro work.
  6. Elaborate on some areas that seem underdeveloped. For example, provide more detail on the immunofluorescence staining results beyond just stating changes in protein expression.
  7. Carefully proofread text and figures. Check for typos, formatting issues, missing scale bars on images, inconsistent punctuation around statistical values, etc.

Discussion

  1. The interpretation and significance of the findings could be expanded on more. Elaborate on the implications for understanding microglial metabolism, neuroinflammation, and hypertension.
  2. Strengthen the logical flow and organization. The discussion jumps around between topics like findings summary, interpretation, previous research, and limitations. Group related ideas.
  3. Reduce repetition from the introduction and results. Avoid restating findings already covered. Focus more on providing context, significance, and future directions.
  4. Add more comparison to previous relevant research. Highlight areas where your findings agree or contrast with past work. Discuss potential reasons for similarities and differences.
  5. Elaborate more on the limitations and alternative interpretations. Comment on things like the choice of aerobic glycolysis assessment techniques, using just one CB2 agonist, etc.
  6. Provide more specifics and citations when stating things like "Many reports show..." to substantiate the points.
  7. Carefully proofread for minor errors in grammar, wording, citations, etc.
  8. End by highlighting the key conclusions enabled by your work, significance for the hypertension field, and important next research steps.

Comments on the Quality of English Language

Carefully proofread for minor errors in grammar, punctuation, spelling and style. For example, ensure consistent verb tense, capitalization of reagents/kits, missing words, etc.

Author Response

Dear Reviewer:

We hope you have a nice day.

We really appreciate the comments and suggestions you provided for the manuscript, which have significantly improved its quality. We have carefully revised the article based on your feedback and made comprehensive changes throughout. We have done our best to revise the article to improve its quality and readability. 

We made the following changes in response to specific questions:

Introduction

We have implemented your suggestions to condense and consolidate information from previous studies regarding the effects of cannabinoids. Additionally, we have provided an explanation of the endocannabinoid system and the rostral ventral lateral medulla (RVLM). The overall flow and logical progression of ideas have been improved, and minor grammatical and stylistic issues have been corrected. Please review the introduction section.

Methods

The organization and structure were improved by reclassifying animal-related experiments as well as cell-related experiments. Specific information on stereotaxic surgical procedures, vendors, catalog numbers, and dilutions of antibodies used for immunoblotting were added. Recurring steps were simplified, and uniform terminology and formatting were used. Specific statistical tests were explained, and minor grammatical, punctuation, spelling, and style errors were corrected.

The supplementary material contains detailed descriptions of all primer sequences for qPCR. Please see Table 1 in the supplementary materials.

Results

We provided explanations for the significance of findings, such as changes in blood pressure and microglia activation status, in the discussion section. We reduced redundancy in the figure legend and added definitions for all acronyms used for the first time. We standardized rules for significant digits, decimal places, and capitalization. We provided more detailed explanations about the results of immunofluorescent staining. We proofread the text and graphs and checked for typos, formatting problems, image scales, and punctuation around statistical values.

There is a question about whether “conscious and anesthetized blood pressure measurements serve different purposes” that needs to be explained to you.

The blood pressure in the awake state typically reflects the normal physiological state. However, it can be affected by many factors and fluctuates greatly. In contrast, inserting the sensor directly into the artery for measurement provides a stable blood pressure value in the anesthesia state. This value can be affected by the depth of anesthesia. So, these two methods can complement each other to provide a more reliable blood pressure reading.

Discussion

We have fully rewritten the discussion section as you have suggested. The first paragraph summarizes the findings of this study. The second paragraph compares previous studies to further affirm that microglia-associated neuroinflammation accompanies the hyperactivity of the PVN area in hypertension. The third paragraph emphasizes the importance of JWH133 activating microglial CB2 receptors. That ultimately leads to relieving neuroinflammation and decreasing blood pressure in angiotensin II-induced hypertensive mice. The fourth paragraph highlights the mechanism involved, which is that activation of the CB2 receptor inhibits aerobic glycolysis. The fifth paragraph points out the shortcomings of the study. Finally, the significance of the study for the field of hypertension and the important next research steps are also noted.

Best Wishes

Reviewer 3 Report

Comments and Suggestions for Authors

The authors described a possible mechanism of hypotension by CB2 activation. The studies are well designed with appopriate assays to delinate the complex downstream signaling of CB2 activation resulting in hypotension. 

The figures apprear to be in low resolution. please correct those. 

Comments on the Quality of English Language

The English language needs to be refined to scientifically cite or describe the experimental results.

A moderate English language editing is needed for the WHOLE manuscript. 

Using first paragraph as an example,

Vague adjectives, such as powerful, strong, should be avoided.

Phrase should be checked, such as, "The CB2 receptor is mainly expressed by immune cells [5] and microglia [6]." Instead of "by", "in" should be used. 

Author Response

Dear Reviewer:

Hope you have a nice day.

Thank you for your helpful comments on improving the manuscript's quality. We have thoroughly revised the manuscript, correcting minor errors in English wording, grammar, punctuation, and formatting.

We made the following changes in response to specific questions:

  1. The images were resized to improve clarity. Please review the revised figures.
  2. The entire manuscript has been edited for the English language. The imprecise language has been revised. Please review the changes in the revised manuscript.

Best Wishes

Round 2

Reviewer 2 Report

Comments and Suggestions for Authors

The authors responded to my comments very well. Thank you.